# Coastal El Niño triggers rapid marine silicate alteration on the seafloor

Sonja Geilert [1] ✉, Daniel A. Frick [2], Dieter Garbe-Schönberg[3,4], Florian Scholz [1], Stefan Sommer[1], Patricia Grasse [1,5], Christoph Vogt[6] & Andrew W. Dale [1]

Marine silicate alteration plays a key role in the global carbon and cation cycles, although the timeframe of this process in response to extreme weather events is poorly understood. Here we investigate surface sediments across the Peruvian margin before and after extreme rainfall and runoff (coastal El Niño) using Ge/Si ratios and laser-ablated solid and pore fluid Si isotopes ($\delta^{30}$Si). Pore fluids following the rainfall show elevated Ge/Si ratios (2.87 µmol mol$^{-1}$) and $\delta^{30}$Si values (3.72‰), which we relate to rapid authigenic clay formation from reactive terrigenous minerals delivered by continental runoff. This study highlights the direct coupling of terrestrial erosion and associated marine sedimentary processes. We show that marine silicate alteration can be rapid and highly dynamic in response to local weather conditions, with a potential impact on marine alkalinity and $CO_2$-cycling on short timescales of weeks to months, and thus element turnover on human time scales.

Marine silicate alteration can broadly be subdivided into two processes, marine silicate weathering and reverse weathering, both involving seawater-particle interactions (Fig. 1)[1–6]. During marine silicate weathering reactive silicates dissolve, thereby consuming $CO_2$ and increasing alkalinity (expressed as bicarbonate $HCO_3^-$, Fig. 1a). This natural $CO_2$ sink and alkalinity source is counteracted by reverse weathering processes, where secondary minerals react with seawater cations and biogenic opal to from cation-rich marine clays[1,4,5,7,8]. In this way, reverse weathering decreases alkalinity and releases $CO_2$ (Fig. 1b)[3,5,9,10]. For a long time, reverse weathering was considered to be a slow process (>10$^4$–10$^5$ years) with sluggish reaction kinetics[3,10–12]. During the last decades though, it was demonstrated that the formation of authigenic clays can occur within months to years, especially in tropical regions with a high secondary mineral supply from terrestrial weathering[1,4]. In this study we highlight that authigenic clay formation can proceed even faster, within weeks to months, following the dissolution of reactive silicates, highlighted here as coupled weathering (Fig. 1c).

Stable silicon (Si) isotopes ($\delta^{30}$Si) can be used as a sensitive tracer for silicate mineral reactions, including coupled mineral dissolution and precipitation reactions, as the light Si isotopes are preferentially taken up in secondary mineral phases[4,11,13–16]. Studies of Si isotopes can be complemented by germanium (Ge) analysis, given that Ge can be regarded as a chemical twin to Si. Germanium is, like Si, delivered to the ocean mainly by rivers and hydrothermal fluids[17,18]. The main sink of Ge and Si isotopes from seawater occurs through uptake by marine organisms, mainly diatoms and sponges[19,20], during adsorption on metal-(oxy)hydroxides and authigenic clay formation in marine sediments[11,13,21–26].

In this study, $\delta^{30}$Si and Si, Ge and Al are analyzed in samples from pore fluids and benthic incubation chambers obtained during a sampling campaign to the Peruvian margin in 2017 (Fig. S1 and Table S1). Shelf station pore fluids are also analyzed during a sampling campaign in 2013 and compared to data from 2008[11]. Sediments are analyzed for bulk geochemistry (major and trace elements), mineral composition (X-ray diffraction (XRD), electron microprobe (EMP)) along with

[1]GEOMAR Helmholtz Centre for Ocean Research Kiel, 24148 Kiel, Germany. [2]GFZ German Research Centre for Geosciences, Section Earth Surface Geochemistry, Telegrafenberg 14473 Potsdam, Germany. [3]Institute of Geosciences, University of Kiel, 24118 Kiel, Germany. [4]Department of Physics and Earth Sciences, Jacobs University Bremen, 28759 Bremen, Germany. [5]German Centre for Integrative Biodiversity Research (iDiv) Halle-Jena-Leipzig, 04103 Leipzig, Germany. [6]Faculty of Geosciences/Crystallography and Geomaterials & MARUM, University of Bremen, 28359 Bremen, Germany. ✉e-mail: sgeilert@geomar.de

Marine silicate alteration

(a)  Marine silicate weathering        (b)  Reverse weathering        (c)  Coupled weathering

**Fig. 1 | Schematic of marine silicate alteration reactions. a** Marine silicate weathering, releasing cations and alkalinity (here $HCO_3^-$) to pore fluids, **b** reverse weathering, consuming cations and alkalinity, and **c** coupled weathering, where the net $CO_2$ and alkalinity turnover depends on the dominance of mineral dissolution versus precipitation. Black symbols represent terrigenous minerals subject to weathering. Note that marine silicate alteration can be traced by $\delta^{30}Si$; marine silicate weathering: release of $^{28}Si$ with a shift in pore fluid $\delta^{30}Si$ to low values (and vice versa for reverse weathering) relative to seawater and/ or diatom $\delta^{30}Si$. Silicon isotopes subject to coupled weathering reactions reflect both mineral dissolution and precipitation, and pore fluid $\delta^{30}Si$ is the net isotope shift ($\Delta_{reaction}$).

laser-ablated stable Si isotope measurements of authigenic clays and bulk sediment. The most striking data originates from the 2017 shelf station that was sampled following a catastrophic rain fall event related to a coastal El Niño that impacted the Peruvian coast[27,28]. The increased delivery of reactive terrigenous silicate particles to the seafloor by increased runoff triggered authigenic clay precipitation. This study highlights the close coupling between continental and marine Si cycles and the potential for large amounts of $CO_2$ to be released to the shallow water column.

## Results and discussion
### Temporal variability of marine silicate alteration at the Peruvian margin
In early 2017, Peru was struck by a coastal El Niño, resulting in heavy rainfall and anomalously warm temperatures, leading to extreme erosion and run-off[27,29–31]. Immediately following this event, pore fluids on the shelf were strongly enriched in heavy Si isotopes, with $\delta^{30}Si$ values up to +3.72 ‰ compared to significant lower values measured during 2013 (+1.33‰; M92) and 2008 (+1.21‰; M77)[11] (Fig. 2; Table S2). The shelf $\delta^{30}Si$ values in 2017 are the highest yet measured in continental margin sediments and can only be explained by extensive authigenic clay formation. In addition, Ge/Si ratios and Al concentrations in the uppermost sediments (<5 cm) were significantly higher in 2017 compared to the earlier sampling campaigns (Fig. 2; Table S2).

Ehlert et al.[11] concluded that the formation of authigenic clays on the Peruvian margin is limited by the delivery of terrigenous matter and thus the availability of Al. This limitation was overcome during the severe rainfall and landslides[27,29], transporting high amounts of terrigenous material to coastal waters[31]. Velazco et al.[31] reported the accumulation of large volumes of terrigenous quartz and feldspars in a newly deposited sediment layer up to 4 cm thick (see their Fig. 3c, d). The Al needed for authigenic clay formation was likely released from these reactive terrigenous minerals, for example albite (Fig. 1c, Table S3), which can dissolve within weeks to months upon contact with seawater[32].

Direct evidence for the formation of authigenic clays in 2017 is provided by EMP analyses of individual amorphous minerals (Fig. 3). Semiquantitative EMP-measurements identified the authigenic clays as being enriched in K (K/Si = 0.22±0.07 (1 SD), $n=11$), Fe (Fe/Si = 0.13±0.08, $n=10$), Mg (Mg/Si = 0.40±0.15, $n=10$) and Al (Al/Si 0.49±0.12, $n=11$) compared to the matrix that was mainly composed of diatoms[33] and terrigenous minerals (Table S3) with lower Al/Si ratios in 2017 (0.09±0.10, $n=5$) and in 2013 (Al/Si = 0.09 ± 0.06, $n=6$) (Fig. 3b,

d, f). The $\delta^{30}Si$ of the authigenic clays ranged between +0.33 ‰ and −0.58 ‰ and was predominantly lower than the bulk sediment in both 2017 (mean +0.03 ± 0.12‰, 1 SD, $n=7$) and 2013 (mean +0.28 ± 0.06 ‰, 1 SD, $n=31$) (Figs. 3 and 4a, Table S4). The authigenic clays showed a trend towards lower $\delta^{30}Si$ with increasing Al/Si ratios, with values intermediate between $\delta^{30}Si$ affected by diatom dissolution and terrestrial and marine authigenic clays, such as smectite[1,34] (Fig. 4a). Reaction transport model results assuming a fractionation factor of $\Delta^{30}Si_{Auth.clay-pore fluid}$ of −3 ‰ (see also Supplement section 3.3) predict a mean $\delta^{30}Si$ of the authigenic clays of −0.56 ‰; overlapping with the lowest measured $\delta^{30}Si$ value of the authigenic clays (Fig. 4a; Table S6). The decreasing $\delta^{30}Si$ values, which are accompanied by increasing Al/Si ratios, are interpreted to result from an increasing maturation of the authigenic clay from an amorphous aluminosilicate precursor to smectite, which is often found as the primary marine authigenic clay phase[1,12].

### Controlling factors on marine silicate alteration reactions
Marine silicate alteration is a complex interplay between mineral/diatom dissolution and the subsequent precipitation of authigenic clays, either being dominated by $CO_2$ consumption and alkalinity release or vice versa or being net neutral when both processes proceed at equal rates (Fig. 1). The combination of Si isotopes and Ge/Si ratios serve as useful tracers to identify these processes[18]. Modern seawater concentrations of Ge and Si show a linear relationship with a slope of $0.76 \times 10^{-6}$ [20], (Fig. 4b). Diatom tests in the modern ocean reflect this ratio[20] and provide a reliable source detection of sedimentary diatom dissolution. The $\delta^{30}Si$ values of pore fluids on the shelf during 2013 in the OMZ and below the OMZ are close to the mean diatom $\delta^{30}Si$ values of Peruvian margin sediments of +0.9 ± 0.6‰[11] (Fig. 2). However, Ge/Si ratios at all stations are above the expected value for diatom dissolution (Figs. 2d and 4b). On the Peruvian shelf in 2017, Ge was strongly enriched compared to previously reported values for similar reducing sediments from the California margin and the Gulf of Mexico[21,22,25,35] (Fig. 4b). Empirical modeling indicates that the enhanced Ge release in 2017 is mainly due to the balance between the dissolution of diatoms and terrigenous material and concurrent authigenic clay precipitation (Fig. 1c; Fig. S4). The increase in Ge with only small changes in Si is apparently caused by slower uptake of Ge into authigenic clay minerals compared to Si (Table S6), also demonstrated by Fernandez et al.[36]. Consequently, Ge is shifted to higher values by diatom and terrigenous material dissolution whereby Si is relatively more quickly taken up in the newly forming authigenic clays at faster rates than Ge.

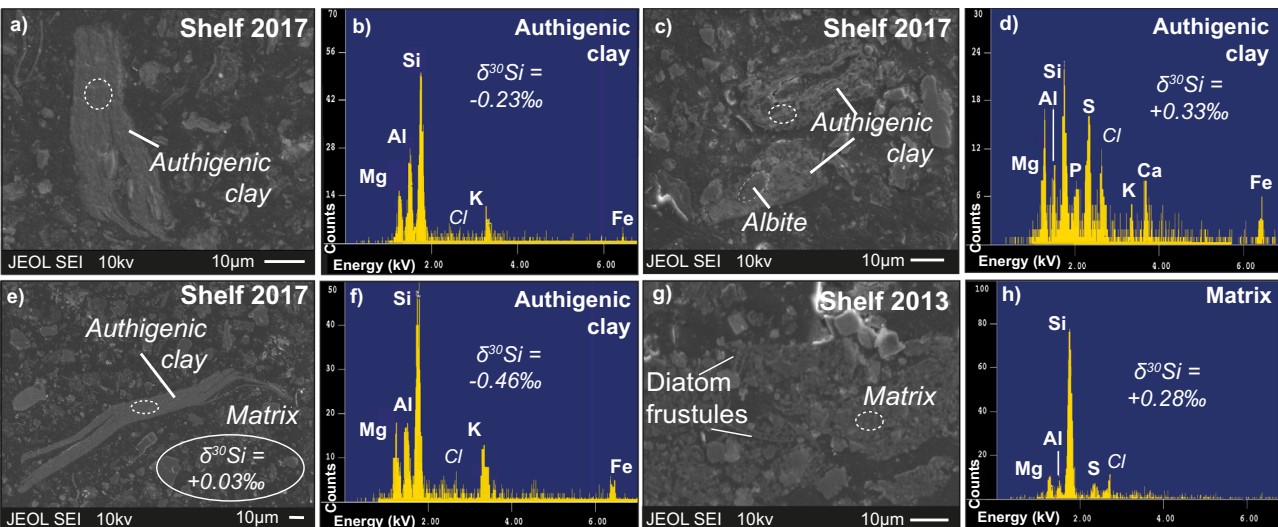

**Fig. 2 | Geochemical variations at the Peruvian margin. a–f** Pore water geochemistry, Ge/Si ratios, Al/Si ratios, and δ³⁰Si values versus depth (centimetres below seafloor; cmbsf). Black horizontal lines denote the sediment-water interface (s-w interface). Values above the s-w interface are measured in bottom water samples. OMZ labels the Oxygen Minimum Zone. Error bars (2 SD of individual measurements) not shown for δ³⁰Si are within symbol size. The δ³⁰Si and Ge/Si ratio for feldspar as representative for reactive silicates are from Savage et al.[38] and Kurtz et al.[24], respectively. The δ³⁰Si and Ge/Si ratio for diatoms are from Ehlert et al.[11] and Sutton et al.[20], respectively. Note the very high δ³⁰Si for the shelf in 2017 indicating extensive authigenic clay formation.

**Fig. 3 | Characteristics of authigenic clays.** Electron microprobe images and semiquantitative mineral analyses of Peruvian shelf samples from 2013 and 2017 (both from 0.5 cmbsf). **a–f** Filmy, veil-like authigenic clays in Peruvian shelf sediments from 2017 are enriched in Si, Al, Mg, K, and with minor amounts of Fe (measurement spots indicated by dashed white ovals). Note the potential authigenic clay and biofilm association in **d** indicated by the presence of P and S, similar to authigenic clays reported for the Ivory Coast-Ghana Marginal Ridge[12]. **g, h** Shelf (2013) sediment matrix composed of diatom frustules and primary minerals. Selected in-situ Si isotope values of the authigenic clays and matrix are shown (note that matrix δ³⁰Si are average values of all measurements and that the complete authigenic clay was ablated for Si isotope analyses, Table S3).

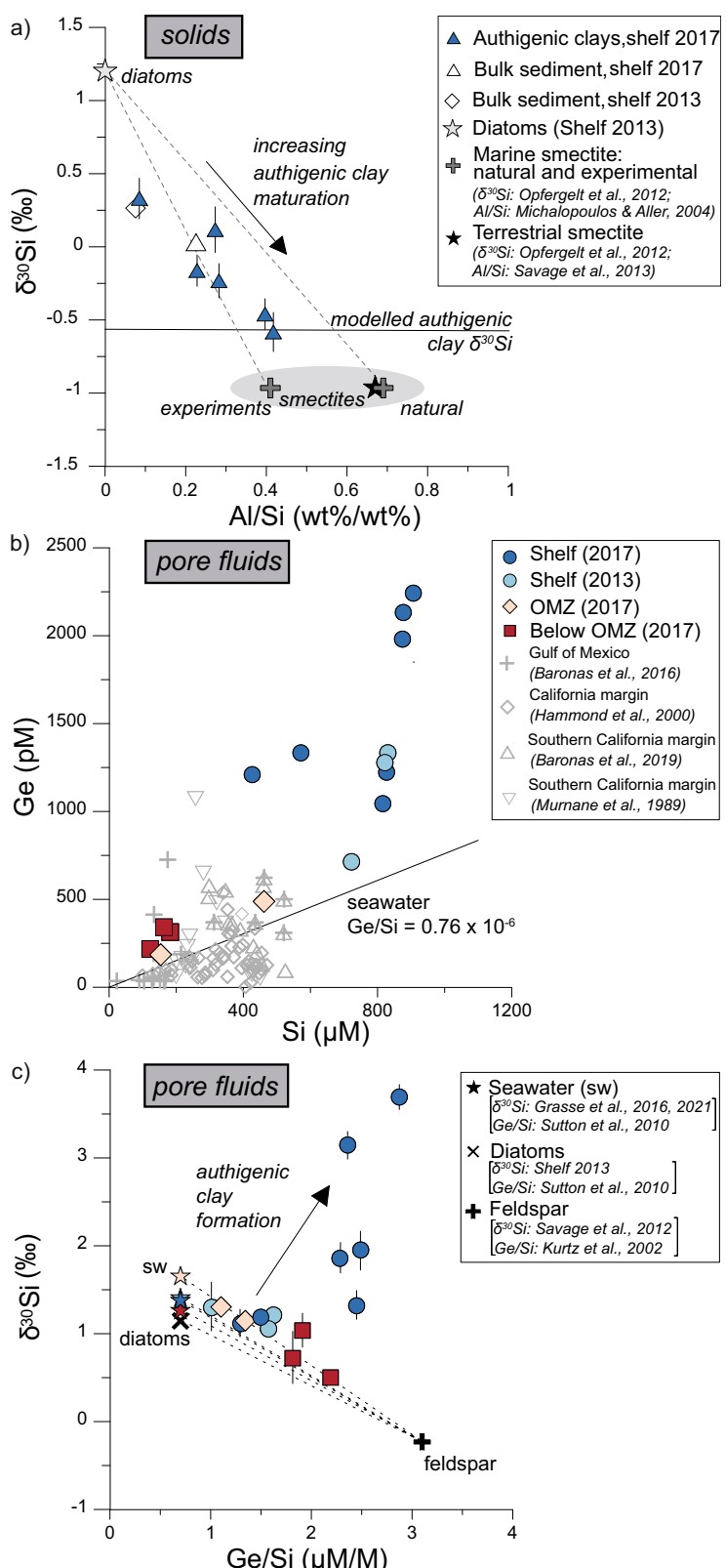

**Fig. 4 | Silicon isotope and geochemical trends in solids and pore fluids. a** In-situ δ³⁰Si of authigenic clays and bulk sediment of the Peruvian shelf in 2017 and bulk sediment analyses from the shelf in 2013. Error bars are 2 SD of individual measurements. Horizontal line is the average modeled authigenic clay δ³⁰Si signature following the coastal El Niño (see also Table S10, Fig. S4). Note that the 'diatom' endmember, which upon dissolution, determines the pore fluid δ³⁰Si from which the authigenic clays precipitate. The trend to lower authigenic clay δ³⁰Si is likely due to increasing maturation of an amorphous gel-like precursor phase to

smectite as described in Baldermann et al.[12]. **b** Ge versus Si concentrations for the Peruvian margin compared to pore fluids from other continental margin areas[21,22,35,56]. **c** Peruvian margin δ³⁰Si versus Ge/Si ratios in pore fluids (symbols as in **b**). Error bars are 2 SD of individual measurements. Most samples fall on mixing lines between dissolving primary silicates (here feldspar), dissolving diatoms and/or seawater (sw). Seawater δ³⁰Si is from Grasse et al.[57,58]. Only samples from the shelf in 2017 deviate from this trend to higher δ³⁰Si and Ge/Si ratios indicating extensive authigenic clay formation (see text).

The sedimentary weathering reactions and the pore fluid Si isotope excursion in 2017 on the shelf are further evident in a plot of $\delta^{30}$Si versus Ge/Si (Fig. 4c). Most Peruvian margin samples fall on mixing lines (equation see supplementary section S4.1) between seawater and pore fluids affected by diatom and feldspar dissolution, respectively. Dissolution of albite (plagioclase feldspar) is the most likely dissolving reactive silicate, since albite is present in high abundance (Table S3), has high Ge concentrations[24] needed to reproduce the measured Ge data in the model (Fig. S4) and has been shown to dissolve in hemipelagic sediments[37]. Albite also has lower $\delta^{30}$Si values[38] and higher Ge/Si ratios[24] compared to diatoms and seawater (Figs. 2d and 4c). Only the samples from the shelf in 2017 deviate from this trend, showing much higher $\delta^{30}$Si values than expected for the measured Ge/Si ratios (Fig. 4c). These high $\delta^{30}$Si values point to extensive authigenic clay formation[11,13–16]. Modeled authigenic clay Si precipitation rates were 291 µmol cm$^{-2}$ yr$^{-1}$ in 2017, whereas in 2013 the rates were significantly lower (13 µmol cm$^{-2}$ yr$^{-1}$; Table S10). Additionally, in 2017 the $\delta^{30}$Si on the shelf inside the benthic chambers deviates from seawater values compared to the other stations (Figs. S2 and S5; Table S5).

## Climate change, extreme events, and marine silicate alteration

Based on pore fluid and sediment geochemical data, we have highlighted the close linkage of riverine mineral supply and marine silicate alteration processes controlling seafloor silicon-carbon turnover. The results of this study indicate that extreme flood events on land can directly influence marine early diagenetic processes by the transport of terrestrial minerals to the marine environment. The data from the Peruvian shelf uniquely illustrate that marine silicate alteration responds rapidly to changing environmental conditions. According to our data, authigenic clay formation as a direct consequence of primary mineral dissolution is a highly dynamic process that reacts almost instantaneously to an increase in terrigenous matter input. Our findings thus challenge the long-standing view on the timeline of authigenic clay formation. It is apparently much faster than previously believed and has the potential to impact marine cation and $CO_2$ cycling on human timescales. Our model of the Peruvian margin shows that, under the present circumstances, reverse weathering is a factor of two higher than weathering of terrestrial minerals (Table S10). Coupled weathering processes are likely widespread on the global seafloor in regions where diverse types of primary and secondary minerals are transported to the ocean.

The frequency of extreme precipitation events triggered by global climate change is increasing (IPCC[39]). Results of this study show that enhanced transport of reactive minerals to coastal waters from land can accelerate and alter marine silicate alteration pathways. While marine silicate weathering impacts seawater chemistry by reducing $CO_2$ and increasing alkalinity[3], the counteracting authigenic clay formation enhances conservation and burial of deposited diatoms in the sediment by alteration to a less soluble phase[4,33] and impacts global element cycles by removing cations from seawater[1,5,40]. Increased delivery of reactive terrigenous minerals to coastal waters by sea-level rise and/ or floods will likely enhance marine silicate alteration and has the potential to significantly impact marine element and $CO_2$ cycling. Further consequences of climate change such as ocean warming, acidification and deoxygenation[41,42] may promote marine silicate weathering rates because low pH, higher temperatures and anoxia in the benthic environment were each found to independently accelerate dissolution reactions[3,43]. The net impact on authigenic clay formation is currently difficult to predict, given that elevated temperatures should increase reverse weathering rates, whereas a lower pH should have the opposite effect or inhibit clay precipitation altogether[9,44].

The response of marine silicate alteration reactions to terrigenous mineral transport at the Peruvian margin is an outstanding example of how climate change might impact marine processes in the future ocean.

## Methods

### Sediment and pore fluid sampling, benthic chamber incubation experiments

Sediment cores up to 40 cm length were retrieved with a multiple corer (MUC). The recovered cores were subsampled under argon for pore fluids and sediment aliquots in the onboard cool room at 4 °C. Bottom water was sampled directly and filtered through a 0.2 µm cellulose acetate filter (Sartorius). The sediment samples were centrifuged to separate the pore fluids from the solids and the supernatant was subsequently filtered (see above). Fluid samples (bottom water and pore fluids) were acidified to pH < 1 with suprapur $HNO_3$, stored in acid-cleaned polyethylene (LDPE) bottles and further analyzed for Si isotopes, Ge and Al concentration at the home laboratory. Sediment samples were freeze-dried for XRD analyses and prepared as thin sections for in situ stable isotope measurements of bulk sediment and authigenic clays.

Benthic landers were deployed on the seafloor in the proximity of the MUC stations and are designed with two circular benthic chambers, described in detail in Sommer et al.[45]. The benthic chambers (internal diameter of 28.8 cm) were partly inserted in the sediment, leaving about 20 to 30 cm water height, resulting in a volume of about 12 to 18 L and covering an area of 651.4 cm$^2$. Prior to incubation, the seawater in the chamber was replaced several times with ambient water to remove suspended particles trapped in the chamber during deployment. The incubations lasted ca.32 h, during which eight consecutive 12 mL samples were collected in acid-cleaned quartz glass tubes. After the incubation, the landers were recovered and the fluids transferred to acid-cleaned LDPE bottles and acidified with distilled $HNO_3$ to a pH < 2. These samples were analyzed for dissolved Si, Ge, Al and Si isotopes.

### Major element analyses of fluids and solids

Sediment samples were freeze dried and ground and subsequently digested in HF (40% Suprapur), $HNO_3$ (Suprapur), and $HClO_4$ (60% p.a.) for major element analyses. The accuracy of the method was tested by method blanks and the reference standards SDO-1 (Devonian Ohio Shale, USGS) and MESS-3 (Marine Sediment Reference Material, Canadian Research Council).

The pore fluid Si concentration and the Al and K concentration of digested bulk solid was measured using the inductively coupled plasma optical emission spectrometry (ICP-OES, VARIAN 720-ES) at GEOMAR. Analytical precision was determined using the IAPSO seawater standard for all chemical analyses[46] and reproducibility was ≤5%.

### Germanium and aluminum analyses of fluids

Elemental concentrations of Al, and Ge in bottom and pore fluid samples were determined by ICP-sector field mass spectrometry (ICP-SF-MS) in high mass resolution mode with ~10,000 R.P. using a Thermo Scientific ELEMENT XR instrument with a PFA micro nebulizer at CAU Kiel. Samples were 1:20 diluted with 2% v/v sub-boiled nitric acid and spiked with yttrium for internal standardization. Six procedural blank solutions were analyzed along with the samples and found to be below limits of quantification of 0.08 mg L$^{-1}$ Si, 0.9 µg L$^{-1}$ Al, and 0.006 µg L$^{-1}$ Ge. Aluminum measured in international certified reference materials (CRM) was as follows: 53.5 µg L$^{-1}$ for synthetic water NIST SRM1640a (certified: 53.0±1.8 µg L$^{-1}$), 130 µg L$^{-1}$ for synthetic water SRM1643f (certified: 133±1.2 µg L$^{-1}$), and 73.7 µg L$^{-1}$ for Thames river water LGC6019 (certified: 73.0±13 µg L$^{-1}$). No certified values are available for Si and Ge in these CRM. Measurement uncertainty (relative) was in the range of 5-10% for all elements.

### Fluid Si isotope analyses

Silicon isotopes were analyzed on a Neptune Plus MC-ICP-MS (Thermo Scientific) and a NuPlasma MC-ICPMS (Nu InstrumentsTM, Wrexham, UK) at GEOMAR. The samples, reference materials (Diatomite,

IRMM-18, Big Batch and an in-house pore fluid matrix standard) and the Si isotope standard materials NBS-28 (bracketing standard) were purified using single column chemistry following Georg et al.[47]. The poly-prep 2 mL column (BioRad, USA) was filled with 1 mL AG® 50W-X8 (200–400 mesh) cation exchange resin. The resin was cleaned before sample loading by sequential washing with 3 M and 6 M HCl, concentrated $HNO_3$, and distilled water. For each sample, an aliquot of the sample solution containing about 4 μg Si was loaded onto a column. Because of the low Si concentration of the benthic incubation experimental samples, theses samples were evaporated before column chemistry to achieve the required Si concentrations. All samples were eluted with 2 mL of distilled water. In order to stabilize the mass bias on the Neptune Plus MC-ICP-MS, the samples, reference materials, and bracketing standard were doped with Mg[48]. After purification and Mg-doping, the sample contained 1 ppm Si for isotope analysis. The typical overall column recovery of Si was greater than 97%.

Silicon isotopes were also measured in medium resolution on the NuPlasma MC-ICPMS using the Cetac Aridus II desolvator. Sample Si concentrations of about 21 μM resulted in a $^{28}Si$ intensity of 3 to 4 V. The distilled water blank was ≤3 mV, resulting in a blank to signal ratio <0.1%. Additionally, silicon isotope measurements were performed at medium resolution on the Neptune Plus MC-ICP-MS. Samples were introduced into the plasma using a 65 μL min$^{-1}$ PFA nebulizer and a PFA spray chamber. The $^{28}Si$ signal was -6.0 V in wet plasma mode and the intensities of $^{28}Si$ in blank solutions were below 20 mV. On both instruments, samples and reference materials were measured at least twice during a sequence and over several days. Results are expressed in the delta notation ($\delta^{30}Si$) as the per mil (‰) deviation from the standard material NBS 28 (Eq. 1).

$$\delta^x Si = \left\{ \frac{(^x Si/^{28}Si)_{sample}}{(^x Si/^{28}Si)_{std}} - 1 \right\}. \quad (1)$$

where $x$ indicates mass 29 or 30. The abbreviation *std* refers to the standard material of NBS-28. On the NuPlasma, reference materials yielded $\delta^{30}Si$ of Big Batch with $-10.6 \pm 0.2$‰ (2 SD; $n = 49$), IRMM018 with $-1.5 \pm 0.2$‰ (2 SD; $n = 48$), Diatomite with $+1.3 \pm 0.2$‰ (2 SD; $n = 44$), and the in-house pore fluid standard of $+1.3 \pm 0.2$‰ (2 SD; $n = 17$). Neptune Plus reference materials yielded IRMM-18 of $1.4 \pm 0.1$‰ (2 SD; $n = 24$); Diatomite of $+1.3 \pm 0.1$‰ (2 SD; $n = 68$), Big Batch of $-10.6 \pm 0.1$‰ (2 SD; $n = 14$) and the in-house pore fluid standard of $0.9 \pm 0.1$‰ (2 SD; $n = 13$). The reference materials measured on both instruments agree well with literature data[49]. All estimates of reproducibility described in this paper are from replicated measurements ($n \geq 4$, 95% confidence limit). Anions were found to affect the Si ionization potential during MC-ICP-MS measurements[48]. Various tests on potential effects of anionic species (Cl$^-$, PO$_3^-$, SO$_4^{2-}$) on Si isotope measurements were undertaken in earlier studies on the NuPlasma[11] and Neptune Plus[50] at GEOMAR. The results show that anion effects are neglectable in the range present in the investigated samples. To further test the comparability of both instruments, several samples were measured on both instruments and $\delta^{30}Si$ values overlap within error (Table S5), ensuring the validity of the Si isotope measurements.

## Solid in situ silicon isotope analyses
In situ silicon isotope ratios were measured by femtosecond laser ablation (fsLA) multi-collector inductively coupled plasma mass spectrometry at the Helmholtz Laboratory for the Geochemistry of the Earth Surface (GFZ Potsdam). The method has been described elsewhere in detail[51,52]. Briefly, a UV (196 nm) laser beam with a pulse length of -150 fs is scanned across the sample, the formed laser aerosol is transported by a stream of helium into an inductively coupled plasma mass spectrometer (Neptune Plus, Thermo Fisher Scientific) where $^{27}Al^+$, $^{28}Si^+$, $^{29}Si^+$ and $^{30}Si^+$ are simultaneously

recorded. The ion optics were operated at medium mass resolution with a typical mass resolving power $m/\Delta m > 5000$ to resolve isobaric interferences (mainly $^{14}N^{16}O^+$ on $^{30}Si^+$). Faraday detectors (equipped with $10^{11}$ Ω amplifiers) were positioned to measure on interference-free, flat-top peak shoulders. The laser frequency was adjusted to obtain ca. 7 V on $^{28}Si$ for both the samples and the bracketing standard NBS28. Results are expressed in the delta notation ($\delta^{30}Si$) as the per mil (‰) deviation from the international δ-zero standard for silicon, NBS 28 (see Eq. 1). The $\delta^{30}Si$ results are presented in Table S4. In every analytical session, three reference materials (BHVO-2, $\delta^{30}Si = -0.27 \pm 0.15$ ‰ (2 s, $n = 11$), GOR132-G, $-0.25 \pm 0.07$ ‰ (2 s, $n = 6$) and ML3B, $-0.26 \pm 0.15$ ‰, (2 s, $n = 7$)) were repeatedly analyzed in between the samples and were in excellent agreement with published values[52,53].

## X-ray diffraction (XRD) analyses
X-ray diffraction (XRD) analyses of the freeze-dried samples were performed at GEOMAR using a Phillips X-Ray diffractometer (PW series) with CoKa radiation and a Ni filter under a voltage of 40 kV and an intensity of 35 mA. The samples were measured in triplicate and the reproducibility of the semiquantitative mineral analysis was ±2%. Quantification of mineral phases of the dried clay samples were based on the freely available X-ray diffraction software MacDiff 4.25[54], Panalytical HighScore™ and the QUAX full-pattern method following Vogt et al.[55]. The standard deviation was ±1–3 % for well-crystallized minerals and ±5 % for the remaining mineral phases.

## Electron microprobe (EMP) analyses
Thin sections of sediments from 0.5 cm sediment depth were investigated for elemental distributions using the Jeol JXA-8200 EMP at GEOMAR. Owing to the amorphous character of biogenic silica and authigenic clays, only semiquantitative measurements were feasible using energy dispersive X-ray spectroscopy (EDS). An acceleration voltage of 15 kV and a beam current of 10 nA were set.

## Reaction-transport modeling
A 1-D reaction-transport model was set up to simulate Si turnover on the Peruvian shelf in 2013 and 2017. The 1-D pore fluid model considers molecular diffusion, bioturbation, sediment burial and compaction, and biogeochemical reactions. The turnover of solids (S) and dissolved pore fluid species (P) was simulated applying the following mass balance equations:

$$d_S \cdot (1 - \Phi) \cdot \frac{\partial S}{\partial t} = \frac{\partial}{\partial x}\left( d_S \cdot (1 - \Phi) \cdot \left( D_B \cdot \frac{\partial S}{\partial t} - w \cdot S \right) \right) + d_S \cdot (1 - \Phi) \cdot R_S. \quad (2)$$

$$\Phi \cdot \frac{\partial P}{\partial t} = \frac{\partial}{\partial x}\left( \Phi \cdot \left( D_S \cdot \frac{\partial P}{\partial t} - v \cdot P \right) \right) + \Phi \cdot R_S. \quad (3)$$

where $S$ is the concentration of solid species in dry sediment, $P$ is the concentration of dissolved species in pore fluids, $t$ is time, $x$ is sediment depth, $d_S$ is the density of dry solids, $\Phi$ is sediment porosity, $D_B$ is the bioturbation coefficient; $w$ is the burial velocity of solids; $R_S$ is the turnover rates of solid species, $R_D$ is the turnover rate of dissolved species, $D_S$ is the molecular diffusion coefficient of solutes in porewater; and $v$ is the burial velocity of pore fluid. The model was set up for five solid species (SiO$_2$ in biogenic opal, SiO$_2$ in authigenic phases, K in sediments, $^{30}$SiO$_2$ in biogenic opal, $^{30}$SiO$_2$ in authigenic phases) and four species dissolved in porewater (H$_4$SiO$_4$, H$_4^{30}$SiO$_4$, K, Ge). The isotopic composition and Ge/Si ratios of the pore fluids ($\delta^{30}Si$) were calculated from the corresponding mole fractions ($^{30}$Si/Si, Ge/Si) applying previously published approaches[11,13]. Full details are given in the supplementary materials.

## Data availability

All data generated or analyzed during this study are included in this published article (and its supplementary information files).

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

## Acknowledgements

We thank the crew of RV Meteor during fieldwork. We are grateful for the support from Antje Beck, Anke Bleyer, Bettina Domeyer, Regina Surberg, Jutta Heinze, and Ulrike Westenströer for their technical and analytical assistance. This study was supported by the Sonderforschungsbereich 754 (Climate-Biogeochemistry Interactions in the Tropical Ocean).

## Author contributions

S.G. conceptualized the study. A.W.D., St.So., F.S., and P.G. sailed the research cruises and conducted the sediment sampling. StSo per-formed the benthic chamber experiments. S.G. and P.G. conducted the fluid Si isotope measurements. D.A.F. analyzed the solid samples for Si isotopes by laser ablation MC-ICPMS analyses. D.G.-S. performed the Ge and Al concentration measurements. S.G. conducted the EMP and XRD analyses. C.V. evaluated the XRD measurements. S.G. and A.W.D. pre-pared the manuscript with contributions from all co-authors.

## Funding

## Competing interests

The authors declare no competing interests.
