## [Peer Review File · Nature Communications]

Coastal El Niño triggers rapid marine silicate alteration on the seafloorReviewer #1 (Remarks to the Author):

- What are the noteworthy results?

It is known that at global scale the cycles of Si and of C are closely linked. Indeed, silicate weathering on land engenders atmospheric CO₂ consumption while reverse weathering engenders CO₂ release in seawater.

Here, the authors show that large scale processes like the El Nino events can have impacts on processes at local scale, especially in specific environments when intense rainfalls can generate extreme erosion and run-off. This article deals with marine silicate alteration which comprises two complementary processes which both involve seawater-particle interactions: silicate weathering and reverse weathering. It is shown that marine silicate alteration can be rapid with CO₂-cycling occurring on short timescales of weeks to months, and that the formation of authigenic clays can occur within weeks to months following the dissolution of reactive silicates. Here, this study highlights occurrence of coupled weathering.

- Will the work be of significance to the field and related fields? How does it compare to the established literature? If the work is not original, please provide relevant references.

The occurrence of reverse weathering in tropical environments like the Amazon delta have been demonstrated already by Michalopoulos et al. (2004). Rahmann et al. (2017) actually showed that this process is active in coastal environments *sensu lato*.

Recently, using in particular variations of delta 30Si signals in abyssal sediments, Luo et al. (2022) showed evidence of active silica diagenesis in deep environments like the Mariana Trench, which is the deepest hadal trench.

The first author has already expertise in Si isotope fractionation in marine pore fluids during early diagenesis and in serpentinization processes (Geilert et al., 2020 a and b). This manuscript goes beyond what has been published so far. Based on variations of geochemical and isotopic signals it gives evidence of both marine silicate weathering and reverse weathering in continental margin sediments, off Peru, at least after El Nino events which trigger massive release of lithogenic materials to the coastal environments.

It is remarkably well illustrated (example : Figure 1) facilitating easy understanding of the main scientific messages for experts and non experts in the marine science field.

- Does the work support the conclusions and claims, or is additional evidence needed?

This manuscript is particularly well argued using a large spectra of techniques through multiparametric approach to support its demonstration. The processes were in particular investigated using Ge/Si ratios and laser-ablated solid and pore fluid Si isotopes ($\delta^{30}\text{Si}$). The pore fluids on the shelf following the extreme rainfall event were characterized by elevated Ge/Si ratios (up to 2.87 $\mu\text{mol mol}^{-1}$) and the highest $\delta^{30}\text{Si}$ values yet measured in continental margin sediments (up to up to 3.72‰). Mineral analyses, numerical modelling of reaction-transport for sediment pore fluids and solids, and simulation of benthic chamber data (cf. Supplementary) suggest that intense silicon isotope fractionation and Ge uptake in the solid phase were related to rapid formation of authigenic clays from reactive terrigenous minerals delivered by extreme erosion and continental runoff.

- Are there any flaws in the data analysis, interpretation and conclusions? - Do these prohibit publication or require revision?

No flaw is detected as regards the data analysis, the interpretation and the conclusions.

- Is the methodology sound? Does the work meet the expected standards in your field?

McKenzie and Garrels (1966) gave examples of reactions which encompass reconstitution and neo-formation of authigenic materials (clays), reactions which are termed « reverse weathering » due to the net consumption of alkalinity and cations (e.g. K⁺, Na⁺, and Mg⁺⁺) and concomitant release of CO₂. Following McKenzie and Garrels (1966) (to be cited), to support the hypothesis that reverse weathering is at work after silicate weathering of material of increased delivery of reactive terrigenous particles to the seafloor by increased runoff triggered by El Nino event, the authors measured variations in alkalinity, delta 30 Si, and Ge. Note that Luo et al (2022) already show the great potential of delta 30Si to evidence reverse weathering, also

hypothesizing that that authigenic clay formation in the bSiO₂-rich sediments of the Mariana Trench is likely governed by the supply of lithogenic materials. I would suggest the authors also refer to this article and write a few sentences to compare the Peruvian and Maria Trench environments.

So, Geilert et al's manuscript is good-science based and corresponds to the best expected standards in marine geochemistry.

- Is there enough detail provided in the methods for the work to be reproduced?

Definitively, yes.

-Conclusive comment

This is a good-science based article on a key topic for marine geochemists, biogeochemists and beyond. It is well written, argued and illustrated. I recommend publication in Nature Communications.

Minor point : references 9 and 25 are identical.

Reviewer #2 (Remarks to the Author):

Geilert et al. have shown some highly interesting observations: substantial elevation of pore water stable silicon isotope composition and Ge/Si ratio at an on-shelf site in 2017, compared to 2013. The authors utilised a suite of geochemical analyses and a reaction transport model to link the above-mentioned pore water observations to very high rates of marine silicate alteration and authigenic silicate formation. I highly commend the amount of high-quality data generated in this study. Analytical precision and accuracy are well reported, and appropriate references have been included. However, I have a few main concerns (expanded below) that the authors need to address, before we can validate the robustness of some of the data interpretation and conclusions.

The limited spatial coverage in the sampling of continental shelf makes it extremely challenging to quantitatively assess the significance of the proposed intense silicate alteration event in 2017. This challenge may be the reason why the authors have been descriptive and qualitative about the importance and implications of the study (Line 140-168), without attempting to quantify the scale and impact of such an event. I understand that often it may be logistically challenging to obtain multiple sediment cores across the continental shelf given other field priorities, but the authors need to clearly address the underlying limitations. One way to overcome the limitation may rely on satellite imagery or hydrographic data from 2017 – the extent of visible turbid, freshwater runoff may indicate the spatial scale of the exceptional terrigenous input due to the El Niño event. Combining the spatial scale information with the computed weathering and reverse weathering rates (Table S10) can then be used to provide a first evaluation of carbon and cation burial or outflux associated with the proposed intense marine silicate alteration event, before the authors could draw robust implications about the significance of such event in impacting carbon and cation cycles (Line 147-149).

Sediment concentration of biogenic opal (bSiO₂) is an important parameter to provide constraint on the reaction transport model (Supplementary information, Section 3.1). Measurements of bSiO₂ were carried out for the 2013 M92 core (Fig. S4), but the analytical procedures have not been described in the manuscript. On the other hand, bSiO₂ measurements were not carried out for the 2017 M136 core (Fig. S4). The authors need to explain how the modelled bSiO₂ for the M136 core is constrained – has an assumption been made that bSiO₂ in the M136 core is similar to that of the M92 core? Is it reasonable that both cores have similar bSiO₂, especially when the EMP images (Fig. 3) show that upper sediment of M136 is dominated by authigenic and terrestrial silicates, contrary to the diatom-dominated M92? Would the unconstrained model-derived bSiO₂ for the M136 core cause significant uncertainties to the main model findings (e.g. Line 133-134)?

Pore water Ge/Si at all sites, including the 2013 M92 on-shelf site, are higher than diatom values (Line 114, Fig. 2d), suggesting that diatom dissolution is not the sole

process that controls the Ge/Si observations. The other processes that can affect pore water Ge/Si are weathering of terrestrial minerals and formation of authigenic silicates (Line 117-119), but these processes are suggested to be inactive (zero rate) for the 2013 M92 on-shelf site (Table S10, Line 134-135). Could the authors provide an explanation to the contrasting results between pore water Ge/Si observations and the model output, which potentially have implications for defining the baseline weathering and reverse weathering rates?

The authors employed a transient modelling approach to examine the intense marine silicate alteration event. I think it will be useful to run the transient model for a longer time period to assess how long the coupled weathering is going to last – 5 years, 10 years or more? Estimated duration of the silicate alteration will contribute to quantifying the scale and impact of such events to carbon and cation cycles (Line 147-149).

Minor comments:

- I think the number of acronyms used in the manuscript can be reduce to accommodate for the broader range of readers. For example, I think marine silicate alteration, marine silicate weathering and reverse weather do not need acronyms.

- Table S6: for readers' convenience, please indicate in the table which parameters are independent observations, cited from other studies, or model-optimised.

- Table S6 and S7: equal sign is confusing. Please use a clearer indication e.g. same as M92.

- Fig. S4: any idea why the model couldn't reproduce the pore water stable Si isotope composition and the Ge/Si at the uppermost depth of the 2017 M136 core? The model-observation difference seems to be significantly large here.

Reviewer #3 (Remarks to the Author):

Reviewer #3 Attachment on the following page

This is a review of the manuscript entitled “Coastal El Niño triggers rapid marine silicate alteration on the seafloor ” by Geilert and others as submitted to Nature Communications.

This is a nice manuscript. Clay mineral authigenesis is historically challenging to fingerprint. This manuscript demonstrates this using silica isotopes and the analysis of Ge in marine sediment and sediment porewater systems. The authors demonstrate that positive silica isotope values and elevated Ge/Si ratios can only be explained by enhanced silicate weathering coupled to reverse weathering. The article is overall clearly written and was simple and a pleasure to review. For me, the data supports the conclusions. To my eye, some tidying up is required. With revisions to the following comments below I can be supportive of publication. – Terry Isson

Fractionation factor. A $\Delta^{30}\text{Si}_{\text{Auth.clay-pore fluid}}$ of -3 ‰ is selected here. This is different from Ehlert et al. (2016) who derives a -2 ‰ value. What is the -3 ‰ here based on?

Ge/Si. From lines 118-123, it sounds like authigenic clay formation increases Ge/Si because clays do not take up Ge/Si in the ratio it is present in. If this is the case, then should the arrow in figure 4 point to the top right of the box?

Reverse weathering is prominent on human time scales. This seems important enough to me that might be worth stating in the abstract.

Net neutral? This is perhaps my main comment. Does reverse weathering perfectly balance silicate weathering in this study (from a silica / alkalinity budget perspective)? It would be nice to see these estimates included.

Alkalinity to silica consumption ratio of reverse weathering. The data is available here to derive this, it could be nice to provide this number.

Line by Line

Line 43: could get rid of the \wedge in 10^5

Line 63: reactive terrigenous material – maybe include the word silicates here ‘reactive terrigenous silicate material’

Line 82: Velazco et al.²⁹ – should the publication year be included?

Line 88: Si_{average} or Si_{av} to be consistent

Line 107-108:

- is it really complex? This process is described really nicely in this paper
- 'consuming Co_2 ... or vice versa' net neutral is an option too as displayed in fig.1

Line 122: 'instantaneously' – is the better term here 'relatively more quickly'

Line 145: 'as a direct consequence of'

Line 147-149: long sentence some commas would be nice

Line 156-157: in what way does authigenic clay formation enhance conservation and burial of deposited diatoms?

Line 163-164: 'authigenic clay formation... difficult to predict' - based on first principles lower pH should lower rates of reverse weathering, and elevated temperature should enhance it – might be worth a mention?

Fig. 3 – Figure is everywhere else spelled out except for this figure.

Fig. 4b – unclear if the grey dots are porewater or if they are seawater

Supplement – would be ideal for all the figures to be together

Table S2 – is cut off

Table S5 – is cut off

References

Ehlert, C., Doering, K., Wallmann, K., Scholz, F., Sommer, S., Grasse, P., Geilert, S., & Frank, M. (2016). Stable silicon isotope signatures of marine pore waters—Biogenic opal dissolution versus authigenic clay mineral formation. *Geochimica et Cosmochimica Acta*, *191*, 102-117.

REVIEWER COMMENTS

Reviewer #1 (Remarks to the Author):

- What are the noteworthy results?

It is known that at global scale the cycles of Si and of C are closely linked. Indeed, silicate weathering on land engenders atmospheric CO₂ consumption while reverse weathering engenders CO₂ release in seawater. Here, the authors show that large scale processes like the El Nino events can have impacts on processes at local scale, especially in specific environments when intense rainfalls can generate extreme erosion and run-off. This article deals with marine silicate alteration which comprises two complementary processes which both involve seawater-particle interactions: silicate weathering and reverse weathering. It is shown that marine silicate alteration can be rapid with CO₂-cycling occurring on short timescales of weeks to months, and that the formation of authigenic clays can occur within weeks to months following the dissolution of reactive silicates. Here, this study highlights occurrence of coupled weathering.

- Will the work be of significance to the field and related fields? How does it compare to the established literature? If the work is not original, please provide relevant references.

The occurrence of reverse weathering in tropical environments like the Amazon delta have been demonstrated already by Michalopoulos et al. (2004). Rahmann et al. (2017) actually showed that this process is active in coastal environments *sensu lato*. Recently, using in particular variations of delta ³⁰Si signals in abyssal sediments, Luo et al. (2022) showed evidence of active silica diagenesis in deep environments like the Mariana Trench, which is the deepest hadal trench.

The first author has already expertise in Si isotope fractionation in marine pore fluids during early diagenesis and in serpentinization processes (Geilert et al., 2020 a and b).

This manuscript goes beyond what has been published so far. Based on variations of geochemical and isotopic signals it gives evidence of both marine silicate weathering and reverse weathering in continental margin sediments, off Peru, at least after El Nino events which trigger massive release of lithogenic materials to the coastal environments.

It is remarkably well illustrated (example : Figure 1) facilitating easy understanding of the main scientific messages for experts and non experts in the marine science field.

- Does the work support the conclusions and claims, or is additional evidence needed?

This manuscript is particularly well argued using a large spectra of techniques through multiparametric approach to support its demonstration. The processes were in particular investigated using Ge/Si ratios and laser-ablated solid and pore fluid Si isotopes ($\delta^{30}\text{Si}$). The pore fluids on the shelf following the extreme rainfall event were characterized by elevated Ge/Si ratios (up to 2.87 $\mu\text{mol mol}^{-1}$) and the highest $\delta^{30}\text{Si}$ values yet measured in continental margin sediments (up to up to 3.72‰). Mineral analyses, numerical modelling of reaction-transport for sediment pore fluids and solids, and simulation of benthic chamber data (cf. Supplementary) suggest that intense silicon isotope fractionation and Ge uptake in the solid phase were related to rapid formation of authigenic clays from reactive terrigenous minerals delivered by extreme erosion and continental runoff.

- Are there any flaws in the data analysis, interpretation and conclusions? - Do these prohibit publication or require revision?

No flaw is detected as regards the data analysis, the interpretation and the conclusions.

- Is the methodology sound? Does the work meet the expected standards in your field?

McKenzie and Garrels (1966) gave examples of reactions which encompass reconstitution and neo-formation of authigenic materials (clays), reactions which are termed « reverse weathering » due to the net consumption of alkalinity and cations (e.g. K⁺, Na⁺, and Mg⁺⁺) and concomitant release of CO₂. Following McKenzie and Garrels (1966) (to be cited), to support the hypothesis that reverse weathering is at work after silicate weathering of material of increased delivery of reactive terrigenous particles to the seafloor by increased runoff triggered by El Nino event, the authors measured variations in alkalinity, delta 30 Si, and Ge. Note that Luo et al (2022) already show the great potential of delta 30Si to evidence reverse weathering, also hypothesizing that that authigenic clay formation in the bSiO₂-rich sediments of the Mariana Trench is likely governed by the supply of lithogenic materials. I would suggest the authors also refer to this article and write a few sentences to compare the Peruvian and Maria Trench environments.

So, Geilert et al's manuscript is good-science based and corresponds to the best expected standards in marine geochemistry.

- Is there enough detail provided in the methods for the work to be reproduced?

Definitively, yes.

-Conclusive comment

This is a good-science based article on a key topic for marine geochemists, biogeochemists and beyond. It is well written, argued and illustrated. I recommend publication in Nature Communications.

Minor point: references 9 and 25 are identical.

We highly appreciate the positive review and validation of our work by this reviewer. The suggested references McKenzie and Garrels (1966) and Luo et al. (2022) were added. We restrained from comparing the Peruvian margin and Mariana Trench, given that the supply of terrigenous minerals and the energy of the environments differ substantially. The doubled references mentioned under 'minor point' were corrected.

Reviewer #2 (Remarks to the Author):

Geilert et al. have shown some highly interesting observations: substantial elevation of pore water stable silicon isotope composition and Ge/Si ratio at an on-shelf site in 2017, compared to 2013. The authors utilised a suite of geochemical analyses and a reaction transport model to link the above-mentioned pore water observations to very high rates of marine silicate alteration and authigenic silicate formation. I highly commend the amount of high-quality data generated in this study. Analytical precision and accuracy are well reported, and appropriate references have been included. However, I have a few main concerns (expanded below) that the authors need to address, before we can validate the robustness of some of the data interpretation and conclusions.

Dear reviewer, thank you for this overall positive review and the appreciation of our analytics. In general, we completely agree with the issues you raised below, though unfortunately, in several cases we could not find a way to implement the suggestions in the manuscript. We hope that you value our efforts and explanation why in some cases the edits could not be conducted. We want to highlight that the main emphasis of our manuscript is the rapidity at which reverse weathering reactions occur, thus challenging the general paradigm that reverse weathering reactions are sluggish. We show that authigenic clay formation can occur within months, thus impacting global cation and carbon cycling on human timescales.

The limited spatial coverage in the sampling of continental shelf makes it extremely challenging to quantitatively assess the significance of the proposed intense silicate alteration event in 2017. This challenge may be the reason why the authors have been descriptive and qualitative about the importance and implications of the study (Line 140-168), without attempting to quantify the scale and impact of such an event. I understand that often it may be logistically challenging to obtain multiple sediment cores across the continental shelf given other field priorities, but the authors need to clearly address the underlying limitations. One way to overcome the limitation may rely on satellite imagery or hydrographic data from 2017 – the extent of visible turbid, freshwater runoff may indicate the spatial scale of the exceptional terrigenous input due to the El Niño event. Combining the spatial scale information with the computed weathering and reverse weathering rates (Table S10) can then be used to provide a first evaluation of carbon and cation burial or outflux associated with the proposed intense marine silicate alteration event, before the authors could draw robust implications about the significance of such event in impacting carbon and cation cycles (Line 147-149).

We highly appreciate this suggestion on how to assess the impact of this weathering event. We examined satellite pictures and hydrographic data but to really be able to judge the extent of this event, the resolution of the data is not sufficient. Also, even if a spatial extent could be estimated, the thickness of the deposited sediment layer and the amount of fresh material deposited will not be possible to extract. Especially the thickness of the deposited sediment layer and the reactive zone of mineral weathering is crucial to estimate the extent of element turnover. Unfortunately, the sampling campaign in 2017 was not designed to cover the spatial distribution of riverine deposits in context of the El Niño event. Therefore, we also regret not having more sediment cores available, as also highlighted by the reviewer. However, our data reveals the rapid timing of marine silicate alteration reactions. This is our main finding in this manuscript, and where we decided to place the emphasis. This is the first study showing the direct coupling of detrital mineral input and related marine silicate alteration. This finding provides new understanding of how Si alteration reactions may play a crucial role in global marine cation and CO₂ turnover on human time scales. The emphasis of follow up studies will be to quantify these element exchanges and the impact on seawater chemistry.

Sediment concentration of biogenic opal (bSiO₂) is an important parameter to provide constraint on the reaction transport model (Supplementary information, Section 3.1). Measurements of bSiO₂ were carried out for the 2013 M92 core (Fig. S4), but the analytical procedures have not been described in the manuscript. On the other hand, bSiO₂ measurements were not carried out for the 2017 M136 core (Fig. S4). The authors need to explain how the modelled bSiO₂ for the M136 core is constrained – has an assumption been made that bSiO₂ in the M136 core is similar to that of the M92 core? Is it reasonable that both cores have similar bSiO₂, especially when the EMP images (Fig. 3) show that upper sediment of M136 is dominated by authigenic and terrestrial silicates, contrary to the diatom-dominated M92? Would the unconstrained model-derived bSiO₂ for the M136 core cause significant uncertainties to the main model findings (e.g. Line 133-134)?

For the simulated M136 core from 2017, the same bSiO₂ rain rate was assumed as for the M92 (2013) core. Though we do not know the exact bSiO₂ content, which might be lower as the reviewer proposed, this variable plays only a minor role in the Si cycle. This is because the rate of silica dissolution is usually not limited by bSiO₂ abundance, especially in diatom-rich sediments in oxygen minimum zones. We can see that this is not a major issue in our data since dissolved Si reaches asymptotic concentrations- suggestive of abundant bSiO₂. Of more significance is the Si isotope

composition of bSiO₂, which was assumed to be constant between the two field campaigns. The Si isotope composition of bSiO₂ is not supposed to be affected by a short-term event such as the El Niño, so that we can reliably use this model input parameter.

Thank you also for highlighting the missing reference for the analytical procedure of bSiO₂. The bSiO₂ data and data for the M92 core were published by Dale et al. (2021)¹, and a reference has been added in the footnote of Table S6.

Pore water Ge/Si at all sites, including the 2013 M92 on-shelf site, are higher than diatom values (Line 114, Fig. 2d), suggesting that diatom dissolution is not the sole process that controls the Ge/Si observations. The other processes that can affect pore water Ge/Si are weathering of terrestrial minerals and formation of authigenic silicates (Line 117-119), but these processes are suggested to be inactive (zero rate) for the 2013 M92 on-shelf site (Table S10, Line 134-135). Could the authors provide an explanation to the contrasting results between pore water Ge/Si observations and the model output, which potentially have implications for defining the baseline weathering and reverse weathering rates?

We agree that the model simulation of the Ge and Ge/Si data could have been better for the 2013 field campaign. We therefore now included terrestrial mineral dissolution and authigenic Si precipitation in the 2013 model scenario in order to fit the Ge data more accurately (Fig. S4). Only low rates of these processes were necessary – being roughly 5 % of the post El Niño rates – which only slightly affected the Si turnover rates during the post El Niño phase (Table S10).

The authors employed a transient modelling approach to examine the intense marine silicate alteration event. I think it will be useful to run the transient model for a longer time period to assess how long the coupled weathering is going to last – 5 years, 10 years or more? Estimated duration of the silicate alteration will contribute to quantifying the scale and impact of such events to carbon and cation cycles (Line 147-149).

In general, we would be in favour of an extension of the modelling time to assess the long-term carbon and cation turnover. However, this is not realistically possible due to (i) having only one sampling point after the El Niño in 2017, and (ii) we do not know exactly how much terrestrial material was added and mixed into the surface layer. In the model, we therefore simulated constant rates of marine silicate alteration to reproduce the geochemical profiles. Without any further empirical time, points we cannot explore the dynamic rate of marine silicate alteration. The informative value of such an exercise would thus be highly speculative, and the subsequent conclusions inaccurate and possibly misleading.

Minor comments:

- I think the number of acronyms used in the manuscript can be reduced to accommodate for the broader range of readers. For example, I think marine silicate alteration, marine silicate weathering and reverse weather do not need acronyms.

The number of acronyms has been reduced as suggested.

- Table S6: for readers' convenience, please indicate in the table which parameters are independent observations, cited from other studies, or model-optimised.

The table was modified accordingly.

- Table S6 and S7: equal sign is confusing. Please use a clearer indication e.g. same as M92.

The equal sign was replaced by the suggested 'same as M92' in Table S7. After modification of the model to account for primary mineral dissolution also during the M92 cruise, as suggested under

'major comments', there are no differences in the model parameter values anymore between M92 and M136 and we included a joint 'Value' column in Table S6.

- Fig. S4: any idea why the model couldn't reproduce the pore water stable Si isotope composition and the Ge/Si at the uppermost depth of the 2017 M136 core? The model-observation difference seems to be significantly large here.

Model results always try to reproduce the general trend of the data. Therefore, deviation between data and models is generally inevitable. To reproduce the very high Si isotope data point at the surface, an even larger isotope fractionation factor would be needed. However, in doing so, the fit to the remaining data points would be compromised. The high Ge/Si ratio is likely due to the very high Ge concentration in the first sample. The overall trend and magnitude of the Ge/Si data is nicely reproduced by the model.

Reviewer 3

This is a review of the manuscript entitled "Coastal El Nino triggers rapid marine silicate alteration on the seafloor" by Geilert and others as submitted to Nature Communications.

This is a nice manuscript. Clay mineral authigenesis is historically challenging to fingerprint. This manuscript demonstrates this using silica isotopes and the analysis of Ge in marine sediment and sediment porewater systems. The authors demonstrate that positive silica isotope values and elevated Ge/Si ratios can only be explained by enhanced silicate weathering coupled to reverse weathering. The article is overall clearly written and was simple and a pleasure to review. For me, the data supports the conclusions. To my eye, some tidying up is required. With revisions to the following comments below I can be supportive of publication. - Terry Isson

Dear Terry Isson, thank you very much for this positive review and the support for our interpretation. Please find our replies to your comments below, which in most cases, could be easily implemented in the manuscript.

Fractionation factor. A $\Delta^{30}\text{Si}_{\text{Auth. clay-pore fluid}}$ of -3‰ is selected here. This is different from Ehlert et al. (2016) who derives a -2‰ value. What is the -3‰ here based on?

The fractionation factor was derived by model iteration. We now included sensitivity tests in the Supplement (section 3.3) to further consolidate this derived fractionation factor. A fractionation factor of -2‰ as derived in the model results by Ehlert et al. (2016)², cannot reasonably reproduce the measured Si concentration and Si isotope data with the current model configuration. A reference to these sensitivity tests is now included on line 100.

Ge/Si. From lines 118-123, it sounds like authigenic clay formation increases Ge/Si because clays do not take up Ge/Si in the ratio it is present in. If this is the case, then should the arrow in figure 4 point to the top right of the box?

We agree that the arrow should be tilted and revised the figure accordingly.

Reverse weathering is prominent on human time scales. This seems important enough to me that might be worth stating in the abstract.

We highly appreciate this comment and happily highlighted the importance for element turnover during human timescales in the abstract in lines 31 -32.

Net neutral? This is perhaps my main comment. Does reverse weathering perfectly balance silicate weathering in this study (from a silica / alkalinity budget perspective)? It would be nice to see these estimates included.

We totally agree that quantification of the alkalinity and CO₂ fluxes would be highly desirable. At present, our model suggests that reverse weathering is a factor of two higher than weathering of terrestrial minerals (Table S10). We included this statement now in lines 151 – 153. We thought about several ways to quantify the element turnover, however the assessment is hampered by the unknown geochemical composition of the authigenic clay product. SEM analyses were only possible on a semi-quantitative basis, given that the still amorphous nature of the clay-precursor did not allow quantitative measurements. Therefore, we can not assess the cation uptake of the authigenic clays at this stage of immaturity. As long as the authigenic clays are still amorphous in nature, quantification of the alkalinity turnover cannot be reliably assessed and would likely require laboratory experiments. We agree, that the next crucial step for marine silicate alteration studies is the quantification of element turnover. Therefore, we concentrated in this study on the rapidity of marine silicate alteration and highlight the importance for cation and CO₂ turnover on human time scales.

Alkalinity to silica consumption ratio of reverse weathering. The data is available here to derive this, it could be nice to provide this number.

Unfortunately, due to the same reasons mentioned in the point before, we can not derive the alkalinity to silica consumption ratios based on our data set. The characterization of the clays was only possible in a semiquantitative way, because of the amorphous character of the authigenic clays. Therefore, we currently do not see a way to derive the TA:Si consumption ratios. We would highly appreciate any suggestions on how to derive these rates based on our data. If we are missing something here, we would be more than happy incorporate the calculation.

Line by Line:

Line 43: could get rid of the ^ in 10⁵
The '^' was removed.

Line 63: reactive terrigenous material – maybe include the word silicates here ‘reactive terrigenous silicate material’
The word ‘silicate’ was added.

Line 82: Velazco et al.29 – should the publication year be included?
The publication year was added, likewise for Ehlert et al. (2016) in line 81 in the revised manuscript.

Line 88: Siaverage or Siav to be consistent
We decided to delete ‘average’ or ‘av’ in general, given that it is obvious from the following mentioned number of analyses and standard deviation.

Line 107-108:

- is it really complex? This process is described really nicely in this paper

We chose the word ‘complex’, because even if the actual processes are rather straightforward, the related cation and CO₂ turnover is difficult to identify and constrain. Also, it is still not understood/quantified which marine silicate alteration process dominates at any particular time, and what exactly controls the switch from dissolution to precipitation and vice versa.

- ‘consuming Co₂... or vice versa’ net neutral is an option too as displayed in fig.1

Thank you for highlighting this important point. We included this third option in the description: ‘Marine silicate alteration is a complex interplay between mineral/diatom dissolution and the

subsequent precipitation of authigenic clays, either being dominated by CO₂ consumption and alkalinity release or vice versa or being net neutral when both processes proceed at equal rates (Fig. 1).’ Lines 108 – 111 in the revised manuscript.

Line 122: ‘instantaneously’ - is the better term here ‘relatively more quickly’
The expression was changed to the proposed one in line 124.

Line 145: ‘as a direct consequence of’
The wording was corrected.

Line 147-149: long sentence some commas would be nice
The sentence was edited accordingly.

Line 156-157: in what way does authigenic clay formation enhance conservation and burial of deposited diatoms?
During authigenic clay formation, diatoms serve as nucleation sites and get altered. The ultimate control on conservation is the solubility, which for altered diatoms is lower compared to unaltered, fresh diatoms (e.g. Michalopoulos & Aller, 2004; Rahman et al., 2017). A short comment was also included in the manuscript in line 161-162.

Line 163-164: ‘*authigenic clay formation... difficult to predict*’ - based on first principles lower pH should lower rates of reverse weathering, and elevated temperature should enhance it - might be worth a mention?
We added a comment on the controls of pH and temperature in lines 168 to 170.

Fig. 3 - Figure is everywhere else spelled out except for this figure.
The spelling was adjusted.

Fig. 4b - unclear if the grey dots are porewater or if they are seawater
The grey symbols are pore fluids. This explanation was added to the caption in line 402.

Supplement - would be ideal for all the figures to be together
We grouped now all the figures and tables of the main text together and all the figures and tables of the transport reaction model.

Table S2 - is cut off, Table S5 - is cut off
The tables were complete, we edited the size to better fit within the page margins.

References

Ehlert, C., Doering, K., Wallmann, K., Scholz, F., Sommer, S., Grasse, P., Geilert, S., & Frank, M. (2016). Stable silicon isotope signatures of marine pore waters—Biogenic opal dissolution versus authigenic clay mineral formation. *Geochimica et Cosmochimica Acta*, 191, 102-117.

Reviewer #2 (Remarks to the Author):

In the revised manuscript and the response to reviewer letter, the authors have addressed my comments, and clearly explained why some of the suggested actions are not feasible. I am generally satisfied with the revision, and I still think the work is original with important results. Data analysis has also been improved. Therefore, I suggest the manuscript is ready for publication.

Reviewer #3 (Remarks to the Author):

Thank you for your responses to my questions. I was thoroughly satisfied with these.